# Protease-resistant streptavidin for interaction proteomics

Mahmoud-Reza Rafiee[1,2,†,‡] (ID), Gianluca Sigismondo[1,2,‡] (ID), Mathias Kalxdorf[1,2], Laura Förster[3], Britta Brügger[3], Julien Béthune[3] (ID) & Jeroen Krijgsveld[1,2,*] (ID)

## Abstract

**Streptavidin-mediated enrichment is a powerful strategy to identify biotinylated biomolecules and their interaction partners; however, intense streptavidin-derived peptides impede protein identification by mass spectrometry. Here, we present an approach to chemically modify streptavidin, thus rendering it resistant to proteolysis by trypsin and LysC. This modification results in over 100-fold reduction of streptavidin contamination and in better coverage of proteins interacting with various biotinylated bait molecules (DNA, protein, and lipid) in an overall simplified workflow.**

**Keywords** chemical derivatization; ChIP-SICAP; contamination; proteomics; streptavidin

**Subject Categories** Methods & Resources; Proteomics

**Mol Syst Biol. (2020) 16: e9370**

## Introduction

The comprehensive characterization of the interactions among biomolecules is a prerequisite to understand the underpinnings of cellular function and regulation. Among the multiple strategies that have been developed to chart such interactions, those based on the capture of a biotinylated bait via streptavidin-coated matrices are particularly powerful as they exploit the very high (near-covalent) affinity between biotin and streptavidin, thus allowing for harsh washing conditions to remove weak (non-specific) interactors. Although this principle has been used for decades, it has found new applications with the advent of proteomic approaches to identify interaction networks more systematically. Two main strategies can be recognized: The first is aimed at the identification of proteins that are biotinylated as a result of their confinement to a cellular location, e.g., the plasma membrane (Kalxdorf *et al*, 2017), or to the direct proximity to a protein of interest conjugated with either a biotin ligase (as in BioID) (Kim *et al*, 2014) or peroxidase (as in APEX) (Rhee *et al*, 2013). In a second approach, biotinylation of a specific class of biomolecules is combined with cross-linking to identify interacting proteins. For example, we have combined ChIP with biotinylation of DNA to identify proteins that co-localize on chromatin with a protein of interest (Rafiee *et al*, 2016). In addition, to identify lipid-binding proteins, bifunctional lipids have been used for photoactivation to induce cross-linking to nearby proteins, followed by lipid biotinylation via click chemistry and subsequent streptavidin enrichment (Haberkant *et al*, 2016).

Although the specificity and binding affinity between streptavidin and biotin is one of the most widely used tools in biochemistry, the difficulty in eluting biotinylated biomolecules from streptavidin resins represents a critical drawback. Harsh elution conditions or direct on-bead digestion promotes the release of streptavidin from the matrix, thus generating a massive contamination of unwanted streptavidin peptides upon protease digestion. As a result, chromatography and subsequent mass spectrometry are readily saturated thus diminishing sensitivity of the analysis. As an alternative, streptavidin with lower affinity for biotin, derivatives of biotin, or cleavable biotin moieties have been proposed; however, these options suffer from decreased specificity, affinity, and solubility issues (Morag *et al*, 1996). The use of an exclusion list may reduce the number of MS/MS events for streptavidin-derived peptides; however, this does not solve the fundamental problem that contamination itself is not prevented, and that adverse consequences for chromatography, peptide identification, and quantification still persist. To overcome these issues, we here introduce a method to render streptavidin resistant to cleavage by trypsin and LysC, thus minimizing contamination of samples with streptavidin-derived peptides. We applied this approach to different use cases and demonstrate that it enhances the sensitivity to identify protein–DNA interactions (in ChIP-SICAP), protein–protein interactions (BioID), and protein–lipid interaction and to profile cell surface proteins (Fig 1A). Furthermore, we show that our strategy to generate protease-resistant streptavidin beads compares favorably to a recent similar approach (Barshop *et al*, 2019). In conclusion, we demonstrate that our protease-resistant beads represent a powerful tool for many applications of MS-based interaction proteomics.

1  Division of Proteomics of Stem Cells and Cancer, German Cancer Research Center (DKFZ), Heidelberg, Germany
2  Medical Faculty, Heidelberg University, Heidelberg, Germany
3  Heidelberg University Biochemistry Center (BZH), Heidelberg, Germany
   *Corresponding author. Tel: +49 6221 421720; E-mail: j.krijgsveld@dkfz.de
   ‡ These authors contributed equally to this work
   †Present address: The Francis Crick Institute, London, UK

# Results

## Chemical modification of streptavidin confers resistance to proteolytic cleavage

We have developed a fast and simple two-step protocol for the chemical modification of lysine and arginine residues of streptavidin via established dimethylation (Hsu *et al*, 2003) and condensation (Patthy & Smith, 1975) chemistries, respectively (Fig 1B). This chemical modification confers resistance to cleavage by trypsin and LysC, the most commonly used proteases in proteomics. The resulting protease-resistant streptavidin (prS) beads allow for direct on-bead digestion and downstream sample processing without any need for further sample fractionation before MS analysis (Fig 1A). Interestingly, prS beads preserve both the enrichment efficiency and the sensitivity of the wild-type streptavidin (wtS) beads (Fig 1C and D), while decreasing the MS intensity of streptavidin peptide contamination 100- to 1,000-fold (Figs 1E and F, and EV1B and C). Here we assessed the benefits of the prS beads in four different

applications to identify (i) chromatin-associated proteins via ChIP-SICAP, (ii) biotinylated proteins through BioID, (iii) biotinylated membrane proteins, and (iv) proteins bound to biotinylated lipids (Fig 1A).

## Protease-resistant streptavidin simplifies and improves performance of ChIP-SICAP

To evaluate the benefits of prS beads for the identification of chromatin-associated proteins, we carried out a ChIP-SICAP experiment by targeting Suz12, one of the core components of the polycomb repressor complex 2 (PRC2) complex (Figs 2 and EV2). Upon Suz12 ChIP, the co-enriched DNA was enzymatically biotinylated and subjected to pull-down with either wtS beads followed by trypsin digestion, or with lysine-modified prS beads followed by consecutive LysC and trypsin digestion (see Materials and Methods for further details). To avoid overloading the LC column by streptavidin peptides coming from the wtS sample, we could inject only 10% of the samples in the single-run MS analyses (Fig EV2A). As expected,

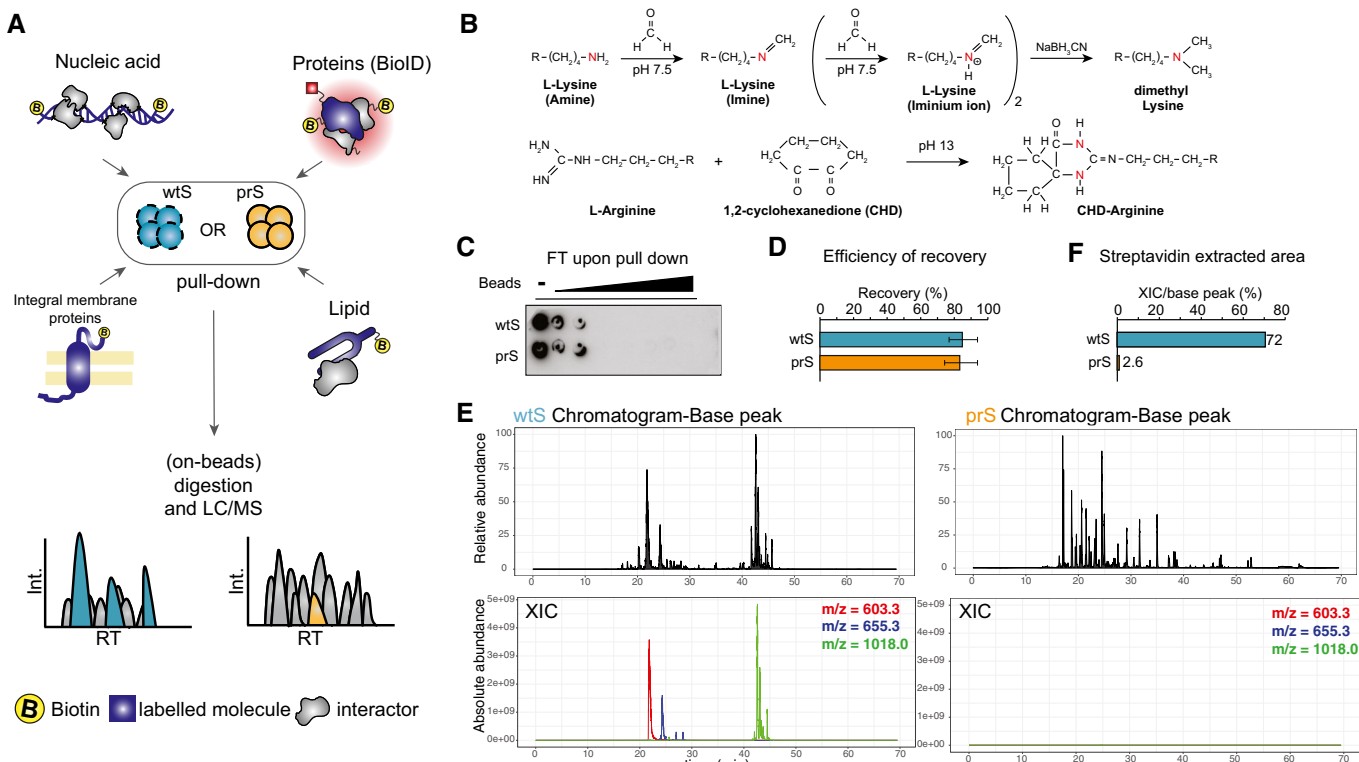

**Figure 1. Protease-resistant streptavidin beads produce dramatically lower streptavidin contaminations in MS compared with regular beads.**

A    Experimental design to target different biotinylated molecules (nucleic acids, proteins labeled by BioID, integral membrane proteins, or lipid-interacting proteins) using either wild-type (wtS) or protease-resistant (prS) streptavidin beads to identify interacting proteins by MS.

B    prS beads were produced by chemical modification of lysines and arginines using dimethylation condensation, respectively.

C    Biotinylated DNA subjected to enrichment with increasing amount of either wtS or prS beads. Flow-through (FT) of the pull-down is probed with anti-biotin antibody.

D    qPCR result shows the efficiency of recovering labeled DNA using either wtS or prS beads. The error bars indicate standard deviations of two replicates.

E    Base peak chromatograms in ChIP-SICAP experiment using wtS (left) or prS (right) beads. Relative abundance is reported on the *y*-axis. The absolute abundance of the top-3 streptavidin peptides is shown using red (*m/z* = 603.3), blue (*m/z* = 655.3) and green (*m/z* = 1018.0).

F    Relative contribution of streptavidin peaks to all base-peaks shown in (E) using either wtS or prS beads.

the wtS sample was highly contaminated with streptavidin peptides and injecting higher amount of the sample was not feasible. In contrast, by using the prS beads, the intensity of streptavidin was reduced from 55% to < 0.1% (Fig 2A). As a result, 39% more proteins (412 versus 296) including many chromatin-related and other nuclear proteins were identified using prS beads in a single injection (Fig 2B). This indicates that reducing streptavidin by prS beads efficiently reveals many otherwise masked proteins. To be able to inject a higher amount of the wtS sample, we subjected the wtS samples to peptide high pH fractionation to disperse streptavidin peptides while extending MS acquisition time for identification of the captured proteins (Fig EV2A). The intensity of streptavidin peptides in fractioned wtS samples ranged between 0.5 and 70% (Fig 2A), and 571 proteins were identified across all fractions

(Fig 2C). This number represents 19% more identified proteins than the single-run experiment using the prS beads; however, this result was obtained by spending 10 times more MS acquisition time. Noteworthy, the abundance of all the core PRC2 components was consistently 5- to 10-fold higher in the single-run prS beads than across the 10 fractions of the wtS sample (Figs 2D and EV2B). In addition, while the overall number of MS/MS spectra was equal, the number of PSMs was even increased in the single-run prS bead experiment compared to wtS sample (Fig 2E).

During the preparation of this work, Barshop *et al* published a similar strategy for streptavidin modification, although using different chemistry (Barshop *et al*, 2019). In a direct comparison of both methods (Fig EV2C), prS beads resulted in 25-fold less streptavidin contamination (ratio of the intensities of the streptavidin

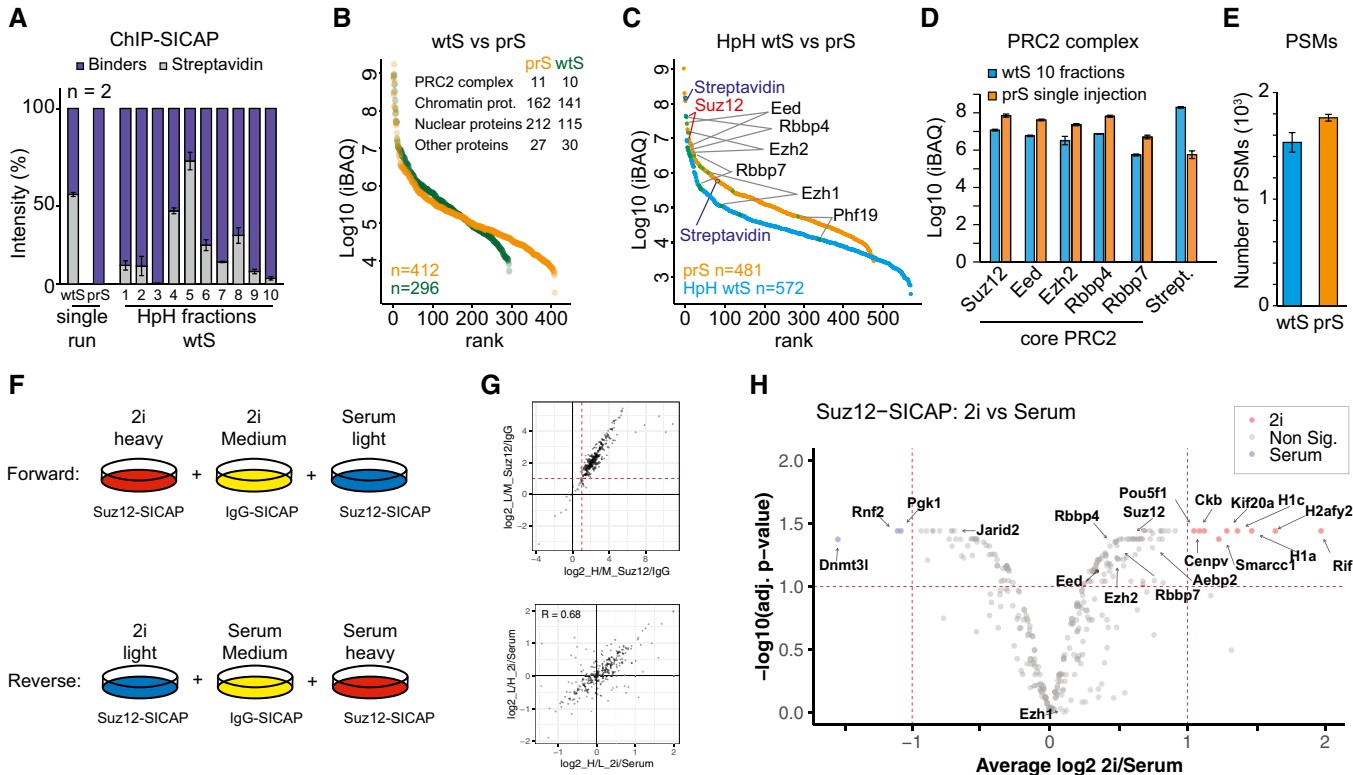

**Figure 2. Studying PRC2 complex by Suz12 ChIP-SICAP using prS beads.**

A  Relative intensity of streptavidin peptides from wtS and prS beads after single-shot MS runs, or from wt beads upon high pH (HpH) fractionation and MS.

B  Intensity-based ranking of proteins identified in single-run MS analysis with either wtS (green) or prS beads (orange). For both experiments, number of identified proteins and their classification is shown.

C  Intensity-based ranking of proteins identified after Suz12 ChIP-SICAP using prS beads and single-injection MS (orange) or using wtS beads followed by MS of HpH-fractionated peptides (blue).

D  iBAQ values of the PRC2 core components after Suz12 ChIP-SICAP, obtained with prS beads and single-injection MS (orange), or with wtS beads and MS of HpH-fractionated peptides.

E  Number of peptide-spectrum matches (PSMs) in HpH wtS or single-injection prS beads after Suz12 ChIP-SICAP.

F  Experimental design for comparing the composition of PRC2 complex by Suz12 ChIP-SICAP in mES cells grown in 2i and serum conditions.

G  Top: scatterplot showing the enrichment of proteins in ChIP-SICAP using a Suz12 antibody compared to IgG as the negative control. > 2-fold enrichment by two replicates was used as the threshold to remove the background. Bottom: scatterplot showing forward and reverse assays of Suz12 ChIP-SICAP.

H  Volcano plot displaying proteins with differential association to Suz12 in 2i and serum conditions as determined using *t*-test statistics. Fold change > 2 and Adj. *P* < 0.1 were used as the threshold.

Data information: Error bars in panels (A, D, and E) indicate standard deviation of two replicates.

peptides; Dataset EV1-comparing with Barshop *et al*—Mock beads). This is due to the absence of streptavidin peptides with $m/z$ 354, 655, and 1,017 which are still liberated from beads modified according to Barshop. By performing ChIP-SICAP with streptavidin beads modified by either procedure, we observed 200-fold less streptavidin contamination and 25% more identification in our prS beads (Fig EV2D and Dataset EV1). Moreover, while almost 2 days (42 h) are required to prepare the beads following Barshop's protocol, 6 h suffice to prepare our prS beads. The better performance of prS beads can most likely be attributed to the difference in reagents and reaction conditions.

Collectively, our data show that prS beads avoid the need for peptide fractionation after ChIP-SICAP, thereby significantly simplifying the overall workflow and saving MS acquisition time, while maintaining or even exceeding the number of peptide identifications. Thus, the use of prS beads in ChIP-SICAP denotes a distinct improvement in the sensitivity and throughput of the method.

## Distinguishing PRC2-associated proteins in primed and ground-state mESCs

The reduction of streptavidin contamination enabled us to perform a triple SILAC experiment to compare proteins co-localized with PRC2 on chromatin in 2i and serum conditions of mouse embryonic stem (mES) cells, which resemble the pre- and post-implantation inner cell mass (ICM) of embryos, respectively. We used normal IgG as the negative control of ChIP-SICAP in the medium channel to evaluate the background proteins, while the heavy and light channels were used for Suz12 ChIP-SICAP. Two biological replicates were carried out for serum and 2i conditions with SILAC label swap (Fig 2F). As a result, we identified 433 proteins using Suz12 ChIP-SICAP, and 292 proteins were quantified using both replicates. In comparison to the normal IgG controls, 262 proteins were enriched by more than twofold in both replicates (Fig 2G and Dataset EV1-comparative Suz12 ChIP-SICAP 2i versus serum). As a threshold for differential co-localization with Suz12, we required a fold change >2 and adj. $P < 0.1$. Consequently, 9 and 3 proteins differentially bind to Suz12 in 2i and serum conditions, respectively (Fig 2H). In addition, the data included all core PRC2 components, almost all of which showed a slight preference to associate with chromatin in 2i (including Suz12, Eed, Ezh2, Rbbp4, Rbbp7, and Aebp2; ~1.5-fold and adj. $P < 0.1$). This observation is in line with previous data reporting that PRC2 binds preferentially to chromatin in the ground state of pluripotency (2i) (van Mierlo *et al*, 2019). Interestingly, of the 16 peptides identified for Aebp2, two peptides specifically belong to isoform 3 and 4 of Aebp2 (Dataset EV1-Aebp2 peptides). Manual inspection of the corresponding MS/MS spectra confirmed the sequence and N-terminus of these peptides starting with an acetylated methionine (Fig EV2E and F), strongly suggesting that indeed this is the isoform-specific protein N-terminus lacking the first 222 amino acids of the canonical Aebp2 sequence. Although on average (across all 16 peptides) Aebp2 is enriched 1.7-fold in 2i conditions in the ChIP-SICAP data (Fig 2H), the peptides derived from the N-terminus of Aebp2 isoform 3/4 are approximately fourfold more abundant (the average of the forward and the reverse experiment, Fig EV2E). To investigate this discrepancy in more detail, we performed a Western blot with a verified monoclonal antibody against Aebp2 using whole-cell lysate of mES cells grown

in 2i or serum conditions (Fig EV2G). We observed two bands in the expected range of Aebp2 isoform 3/4 in the 2i condition (band A and B, Fig EV2G), while in the serum condition we mainly observed only one band (band A). Importantly, this dual-band pattern has been previously observed and confirmed to originate from Aebp2 (Lee *et al*, 2018). Interestingly, band B is ~3.8-fold more intense in 2i, which is perfectly in line with our MS results (Fig EV2E and F), while band A is ~1.4-fold more intense, matching with the overall abundance difference of Suz12-associated Aebp2 on chromatin (Fig 2H). These results suggest that band B represents the full isoform 3/4 and that its expression is induced in 2i conditions. In addition, the longer isoform is likely further processed into the lower band (Band A), potentially by N-terminal processing. Since the ratios of band A and B in the total proteome are in direct agreement with MS-derived ratios in ChIP-SICAP, this indicates that both variants associate with Suz12 in a non-selective manner. Whether these two forms of Aebp2 fulfill distinct functions, within or outside the PRC2 complex, remains to be established.

In contrast to the other PRC2 components, Jarid2 co-localized with Suz12 more in serum condition (~1.5-fold and adj. $P < 0.1$) (Fig 2H). Previous studies have shown that Jarid2 functions to recruit PRC2 and to modulate its histone methyltransferase activity (Li *et al*, 2010; Pasini *et al*, 2010). Increased co-localization of Jarid2 with PRC2 in serum condition highlights its role in PRC2 targeting and establishing the poised pluripotent state in serum condition.

Among the proteins that preferentially bind to PRC2 in serum condition, we observed Rnf2 (Ring1b or RING2), an E3 ubiquitin ligase that adds a ubiquitin moiety to H2A on K119 (Fig 2H). Rnf2 is an essential component of PRC1, a complex that is closely interconnected with PRC2 (Cao *et al*, 2005; Vidal & Starowicz, 2017) and required to establish repressive chromatin state on many developmental genes (Taherbhoy *et al*, 2015). Furthermore, we observed Dnmt3l preferentially binds to PRC2 in the serum condition. Dnmt3l is a catalytically inactive DNA methyltransferase regulating both Dnmt3a and Dnm3b, and reported to interact and "sequester" Ezh2 from the interaction with Dnmt3a and Dnmt3b, thus maintaining a low DNA methylation of H3K27me3 loci (Neri *et al*, 2013).

Conversely, in 2i condition we observed a significant higher association of PRC2 with both histone macro-H2A (H2afy2) and the linker histones, H1a and H1c. Considering the repressive function of these histones, our results suggest a plausible mechanism how PRC2 may suppress gene expression in 2i condition. Moreover, Rif1, a protein important for the stabilization of H3K9 methylation and telomere homeostasis (Dan *et al*, 2014), showed a significant binding with PRC2 in 2i condition. Rif1 was also reported to promote DNA silencing in the absence of DNA methylation by promoting the deposition of H3K27me3 (Li *et al*, 2017). Interestingly, we observed significant association of PRC2 with Pou5f1 (Oct4) and Smarcc1 (BAF155, a member of npBAF and nBAF) predominantly in 2i condition, in line with our previous report on co-localization of Oct4, Suz12, and Smarcc1 in mES cells (Rafiee *et al*, 2016). Altogether, the decreased streptavidin contamination resulting from the use of prS beads allowed us to design a more advanced ChIP-SICAP workflow with less sample handling, decreased experimental timing, and better accuracy, to identify changes in chromatin-associated protein interactions during cell fate transition.

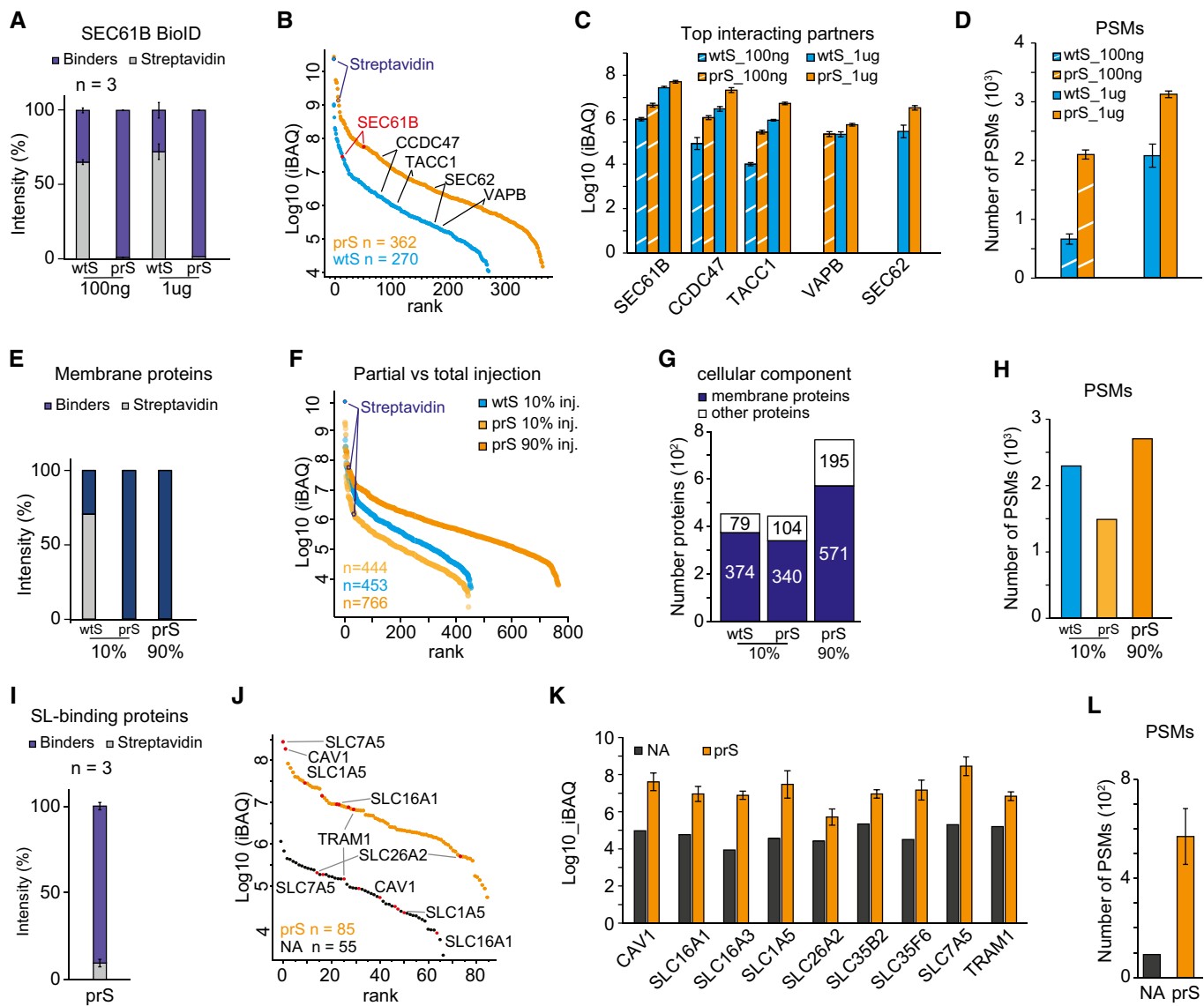

**Figure 3. Streptavidin depletion using prS beads allows for better enrichment and identification of biotinylated targets and their interactors.**

A–D  SEC61B BioID. (A) Relative intensity of streptavidin peptides in 100 ng or 1ug BioID experiments performed with either wtS or prS beads. The negative control only expressed BirA*. (B) Intensity-based ranking of identified proteins in 1ug experiment. (C) Intensity of SEC61B and its top interacting proteins in 100 ng or 1ug BioID experiments using either wtS or prS beads. (D) Number of PSMs in BioID experiments.

E–H  Integral membrane proteins. (E) Percentage of streptavidin peptides after enrichment of biotinylated membrane proteins, identified in 10% of wtS or prS sample, and in the residual 90% of the prS sample. (F) Intensity-based ranking of proteins identified in the experiments in (E). (G) Number of total and membrane proteins identified in the experiments in (E). (H) Number of PSMs in the experiments in (E).

I–L  Sphingolipid (SL)-binding proteins. (I) Relative intensity of streptavidin after lipid enrichment in prS beads. (J) Intensity-based ranking of proteins identified using either NeutrAvidin or prS beads. (K) Intensity of common sphingolipid-binding proteins identified after enrichment via NeutrAvidin (NA, black) and prS beads (orange). (L) Number of PSMs in the respective experiments in (K).

Data information: Error bars in panels (A, C, D, I, K, and L) indicate standard deviation of three replicates.

## Protease-resistant streptavidin enhances coverage in BioID

To evaluate the advantage of the prS beads in protein–protein interaction studies, we used it in BioID, an approach that is designed to chart protein networks by capturing proteins that are biotinylated as a result of being in the vicinity of a protein of interest that is fused to a biotin ligase (Roux *et al*, 2012). Specifically, we were interested to identify interaction partners of SEC61B, a subunit of the ER-localized SEC61 translocon channel.

BioID was carried out in HeLa cells expressing mutant BirA (BirA*) fused to SEC61B or in wild-type cells as negative control that expressed only BirA* (Figs 3A–D and EV3A). The experiments

were performed in triplicate, and interacting proteins were identified in a single-shot MS analysis upon on-bead digestion. For each experiment, two different amounts of starting material were injected corresponding to either a low amount (100 ng) or high amount (1 μg) of sample. When using wtS beads, in both cases more than 60% of the overall sample intensity was attributed to streptavidin peptides, while this dropped to < 2% when using prS beads (Figs 3A and EV3B and C). The intensity of the bait protein SEC61B was fivefold higher when prS beads were used (Fig 3B and Dataset EV2). In addition, in the prS-treated sample 362 proteins were identified compared to 270 with wtS beads, with an overall ~10-fold higher intensity (Fig 3B), as exemplarily shown for four known interacting partners (Fig 3C). Furthermore, the overall number of PSMs was ~1.5-fold higher when using prS beads (Fig 3D), mirrored by a higher number of both MS/MS in prS compared with wtS bead BioID experiments, demonstrating greatly enhanced protein identification (Fig EV3D). The increase in intensity and PSMs was most prominent in low-input (100 ng) samples, most likely because especially low-abundance peptides are easily masked by intense streptavidin peptides eluted from wtS beads.

We evaluated the intensity of the SEC61B protein and 19 additional proteins that were also found in another BioID experiment of SEC61B performed in HEK-293 cells (preprint: Go et al, 2019). Interestingly, by injecting the same peptide amount, we observed for 18 out of these 20 proteins a statistically significant difference in intensity in prS beads over wtS beads (Dataset EV2: the shortlist). SRP14 and SRP72, two known subunits of the SRP, were detected with higher intensity in prS beads over wtS bead BioID experiments (Dataset EV2 SEC61B-BioID). Taken together, these results indicate that BioID with prS beads identifies more PSMs and proteins with enhanced intensity, leading to a better identification of the bait protein together with known as well as novel interacting partners.

### Deeper characterization of cell surface proteins with protease-resistant beads

We also compared the performances of wtS and prS beads in characterizing surface-exposed integral membrane proteins in HeLa cells, achieved via biotinylation of their sugar moiety (Kalxdorf et al, 2017) (Figs 3E–H and EV3E). Initially, after enrichment on wtS beads, 10% of each sample was injected for LC-MS to avoid overloading the analytical column by intense streptavidin peptides. Indeed, streptavidin comprises ~70% of the total protein intensity in the wtS sample (intensity: > 1e11), compared to < 1% (intensity: 9.4E6) in the equivalent sample from prS beads (Figs 3E and EV3F). Importantly, this reduction of streptavidin contamination allowed us to analyze the remaining 90% of the prS sample without saturating chromatography. This raised the number of identified proteins from 444 to 767 at overall higher iBAQ intensities (Fig 3F and G and

Dataset EV3), which included 68% more membrane proteins (571 versus 340). Furthermore, the number of PSMs was significantly higher compared to the 10%-injections (Fig 3H). These data demonstrate that the reduction of streptavidin contamination allows to inject a larger fraction of the samples, thus boosting the number of PSMs and protein identifications.

### Characterization of lipid-binding proteins

We then evaluated how prS beads perform in the enrichment of lipid-binding proteins (LBP, Fig 3I–L). For this purpose, we used engineered A594 cells in which bifunctional sphingolipids (SL) can be first cross-linked to nearby proteins by photoactivation and then biotinylated through click chemistry (Gerl et al, 2016) (Fig EV3G and H). The LC-MS analysis of LBPs captured and digested on prS beads indicated that the intensity of streptavidin peptides accounted for < 10% of the total peptides intensity (Fig 3I). When comparing prS beads to high-capacity avidin beads, commonly used for the identification of LBPs (Trajkovic et al, 2008), the prS beads identified more proteins (85 versus 65) with consistently 100-fold increase in iBAQ intensities (Fig 3J). Focusing on sphingolipid-binding proteins identified by both bead types, we observed a 10- to 1,000-fold higher protein intensity in the prS bead experiment (Fig 3K) and overall sixfold increase in PSMs (Fig 3L) compared with the NeutrAvidin pull-down. Taken together, these data demonstrate that the use of prS beads is highly advantageous also for the enrichment of biotinylated lipids, increasing the sensitivity for the identification of LBPs.

## Discussion

Collectively, our data show how protease-resistant streptavidin offers distinct advantages to characterize protein networks across a wide range of applications. This arises from the 100- to 1000-fold decrease in streptavidin contamination, thereby rendering low-abundance peptides detectable for identification by MS. The strategy offers the choice for the on-bead chemical modification of only lysine, or of both lysine and arginine residues, to generate LysC- or trypsin-resistant streptavidin, respectively. Either way, prS beads allow for the on-bead digestion of captured proteins and direct downstream analysis by LC-MS. This avoids contamination by streptavidin-derived peptides and circumvents the need for peptide fractionation, thereby reducing sample handling and MS analysis time and, most importantly, leading to overall better sensitivity and increased sampling depth. In conclusion, we expect that protease-resistant streptavidin beads will find broad use in the characterization of protein networks in the many flavors of biotin-based enrichment strategies that have been developed over decades.

# Materials and Methods

## Reagents and Tools table

| Reagent/resource | Reference or source | Identifier or catalogue number |
|---|---|---|
| **Cell lines** | | |
| 46C embryonic stem cell (*M. musculus*) | The laboratory of Austin Smith | |
| HeLa cell (*H. sapiens*) | Kalxdorf *et al* (2017) | |
| HeLa cell (*H. sapiens*) | Schopp *et al* (2017) | Cell lines stably express SEC61B-BirA* fusion protein or BirA*-expressing cells |
| A549 cells (*H. sapiens*) | Gerl *et al* (2016) | ΔSGPL1 |
| **Antibodies** | | |
| Rabbit anti-Suz12 | Cell signalling tech | D39F6 |
| Rabbit anti-Aebp2 | Cell signalling tech | 14129 |
| Rabbit anti-biotin | Cell signalling tech | 5597 |
| Goat anti-rabbit HRP | Cell signalling tech | 7074 |
| Goat anti-rabbit HRP | Santa Cruz | sc-2357 |
| Normal rabbit IgG | Cell signalling tech | 2729 |
| **Oligonucleotides** | | |
| ActB-Intron3-F | AAAGCCACAAGAAACACTCAG | |
| ActB-Intron3-R | TATTGAGTAGATGCACAGTAGGTC | |
| **Chemical, enzymes and other reagents** | | |
| Streptavidin magnetic beads | NEB | S1420S |
| NeutrAvidin beads | Thermo Scientific | 29200 |
| AMPure XP beads | Beckman Coulter | A63881 |
| NaOH (10M) | Sigma | 72068-100ML |
| 1,2-cyclohexanedione | Sigma | C101400-1G |
| PBS-T | PBS + Tween 0.1% | |
| PBS-T, pH 13 | add NaOH to PBS to increase the pH to 13. This should be prepared freshly | |
| Sodium cyanoborohydride (NaBH$_3$CN) | Sigma | 8180530010 |
| Formaldehyde 16% | Pierce | 28908 |
| Dynabeads Protein A | Thermo Scientific | 10002D |
| Complete protease inhibitor | Roche applied science | 11 836 153 001 |
| Terminal deoxynucleotidyl transferase (TdT) | Thermo Scientific | EP0162 |
| Biotin-ddUTP | Jena Bioscience | NU-1619-BIOX-L |
| Biotin-dCTP | Jena Bioscience | NU-809-BIOX-S |
| Biotin-7dATP | Jena Bioscience | NU-835-BIO-S |
| BCA Protein Assay Kit | Thermo Scientific | 23225 |
| Klenow exo- | NEB | M0212S |
| dNTP solution set | NEB | N0446S |
| T4 PNK | NEB | NEB, M0201S |
| ZipTip Pipette Tips | Millipore | ZTC18S096 |
| Acetonitrile (anhydrous, 99.8%) | Sigma | 271004 |
| RNase A | Thermo Scientific | EN0531 |
| Dithiotritol (DTT) | Sigma | D0632 |
| Iodoacetamide (IAA) | Sigma | I1149-5G |
| Ammonium bicarbonate (Ambic) | Sigma | 09830-500G |
| ECL Western Blotting Detection Reagents | GE Healthcare | RPN2106 |

**Reagents and Tools table** (continued)

| Reagent/resource | Reference or source | Identifier or catalogue number |
|---|---|---|
| SYBR Premix DimerEraser | Takara Bio | RR091B |
| SP3 beads (Sera-Mag magnetic beads) | GE Healthcare | 65152105050250 and 45152105050250 |
| $^{12}C_6$ $^{14}N_4$ L-arginine | Silantes | 201003902 |
| $^{12}C_6$ $^{14}N_2$ L-lysine | Silantes | 211003902 |
| $^{13}C_6$ $^{14}N_4$ L-arginine | Silantes | 201203902 |
| $^{12}C_6$ $^{14}N_2$-d4 L-lysine | Silantes | 211103913 |
| $^{13}C_6$ $^{15}N_4$ L-arginine | Silantes | 201603902 |
| $^{13}C_6$ $^{15}N_2$ L-lysine | Silantes | 211603902 |
| **Software** | | |
| MaxQuant 1.5.2.8 or 1.6.2.6 | https://www.maxquant.org | |
| Proteome Discoverer 2.2 | https://www.thermofisher.com/order/catalog/product/OPTON-30812 | |
| Fiji | https://imagej.net/Welcome | |
| R studio, R | https://rstudio.com | |
| **Other** | | |
| Thermo Orbitrap Velos | Thermo Scientific | |
| Thermo Orbitrap Fusion | Thermo Scientific | |
| Thermo Easy-nLC 1200 | Thermo Scientific | |
| Dionex UltiMate 3000 LC System | Dionex | |
| Pico Bioruptor | Diagenode | |
| Amersham Imager 680 blot and gel imager | GE Healthcare | |
| Trans-Blot Turbo Transfer | Bio-Rad | |
| ChemiDoc | Bio-Rad | |

## Methods and Protocols

Generation of protease-resistant streptavidin beads through two-step chemical modification.

Required material:

- Streptavidin magnetic beads (e.g., NEB S1420S).
- NaOH (10M) Sigma 72068-100ML.
- 1,2-cyclohexanedione Sigma C101400-1G.
- PBS-T: PBS + Tween 0.1%.
- PBS-T, pH 13: add NaOH to PBS to increase the pH to 13. This should be prepared freshly.
- Sodium cyanoborohydride ($NaBH_3CN$) Sigma 8180530010.
- Formaldehyde 16% (Pierce 28908).
- Reagent A: Prepare 8 ml Formaldehyde 4% in PBS-T (v/v).
- Reagent B: Prepare 8 ml Sodium cyanoborohydride 0.2 M in PBS-T.
- Reagent CHD: dissolve 120 mg of cyclohexanedione in 14 ml PBS-T, pH 13.

Note: reagent A and B are toxic. You should prepare them in a fume hood and discard the wastes properly.

The protocol below allows for the production of streptavidin beads resistant to LysC and/or trypsin digestion. For LysC-resistant beads, skip step 5 to step 9 and continue from step 10.

1  Pour 5 ml of streptavidin beads (e.g., NEB S1420S) into a 15 ml tube
2  Put the tube on the magnet, wait 5 min and then discard the liquid
3  Wash the beads with 10 ml of PBS-T
4  Put the tube on the magnet, wait 5 min and then discard the liquid
5  Resuspend the beads in 14 ml of reagent CHD
6  Rotate at room temperature for 4 h
7  Put the tube on the magnet, wait 5 min and then discard the liquid
8  Wash the beads with 10 ml PBS-T
9  Put the tube on the magnet, wait 5 min and then discard the liquid
10  Resuspend the beads in 7 ml of reagent A
11  Add 7 ml of reagent B
12  Rotate 2 h at room temperature
13  Put the tube on the magnet, wait 5 min and then discard the liquid properly
14  Wash the beads with 10 ml Tris–HCl 0.1M pH 7.5
15  Put the tube on the magnet, wait 5 min and then discard the liquid properly
16  Wash the beads with PBS-T
17  Put the tube on the magnet, wait 5 min and then discard the liquid properly
18  Resuspend the beads in 5 ml PBS-T
19  Keep the beads in the fridge. The beads are stable at +4°C for months

### SEC61B interactors investigated through BioID

BioID experiments were performed in stable HeLa cell lines expressing SEC61B-BirA* fusion protein or BirA*-expressing cells, as control, as described previously (Schopp *et al*, 2017) with minor modifications. Briefly, around $30 \times 10^6$ cells per experiment were lysed in 1 ml of lysis buffer (Tris–HCl pH 7.4 50 mM, NaCl 500 mM, SDS 0.4%, EDTA 5 mM, DTT 1 mM, 1× Protease Inhibitor Cocktail from (Roche)), then mechanically disrupted by 10 passages through a 25G needle and sonicated for 4 cycles of 30 s ON/30 s OFF in a Pico Bioruptor (Diagenode). Triton X-100 was added to a final 2% concentration, and the concentration of NaCl was adjusted to 150 mM final with Tris–HCl pH7.6 50 mM. Upon centrifugation at 4°C 16,000 *g*, the supernatant was incubated with 200 μl of equilibrated prS or wt beads o/n. at 4°C on rotating wheel. Beads were then recovered on a magnetic rack and washed sequentially twice with washing buffer 1 (SDS 2% in water), twice with washing buffer 2 (HEPES 50 mM pH 7.5, Na-deoxycholate 0.1%, Triton X-100 1%, NaCl 500 mM, EDTA 1 mM), twice with washing buffer 3 (Tris–HCl pH 8 10 mM, NP-40 0.5%, Na-deoxycholate 0.1%, LiCl 250 mM, EDTA 1 mM), and twice with washing buffer 4 (Tris–HCl pH 7.4, NP-40 0.1%, NaCl 50 mM). Beads were then conditioned with 500 μl of ammonium bicarbonate (AmBic) 50 mM for 5 min at RT and resuspended in 28 μl of SDS 0.1% in AmBic 50 mM. Protein disulfide bonds were reduced and alkylated with DTT 7 mM at 50°C for 30 min and iodoacetamide 20 mM at RT for 45 min in the dark, respectively. Upon quenching with DTT, 300 ng of trypsin was added (Promega V5111) for overnight digestion at 37°C with 700 rpm shaking. Peptides contained in the supernatant were subjected to de-salting by SP3 beads protocol as previously described (Hughes *et al*, 2014, 2019; Rafiee *et al*, 2016). Peptides were eluted in trifluoroacetic acid (TFA) 0.1% and then quantified with quantitative colorimetric peptide assay (Pierce, 23275). A peptide amount corresponding to either 100 ng or 1 μg was loaded on a trap column (PepMap100 C18 Nano-Trap 2 cm × 100 μm) followed by separation over a 25 cm analytical column (nanoEase MZ Peptide BEH C18 column, 130 Å, 1.7 μm, 75 μm) using a 70 min linear gradient of acetonitrile from 6 to 40% (Thermo Easy-nLC 1200, Thermo Fisher Scientific). Peptides were analyzed on a Tri-Hybrid Orbitrap Fusion mass spectrometer (Thermo Fisher Scientific) operated in data-dependent acquisition mode with HCD fragmentation. The MS1 and MS2 scans were acquired in the Orbitrap and ion trap, respectively.

### Chromatin-associated interactors of Suz12 investigated through ChIP-SICAP

ChIP-SICAP experiment was performed as already described in (Rafiee *et al*, 2016) with some modifications. A version of the updated protocol is maintained at protocols.io: dx.doi.org/10.17504/protocols.io.bcrriv56.

Briefly, 46C cell pellets corresponding to $24 \times 10^6$ cells were cross-linked with 1.5% formaldehyde final concentration and were resuspended in 5.5 ml Tris–HCl 10 mM pH 8. After 5 min on ice, 0.5 Triton X-100 10% was added to the samples. After 10 min, the cells were spun at 1,000 *g* for 2 min to precipitate the cells. Then, the cells were resuspended in lysis buffer 3 (Tris–HCl pH8 10 mM, Na-deoxycholate 0.1%, Na-lauroylsarcosine 0.5%, NaCl 100 mM, EDTA 1 mM), and the cells were transferred to 6× 1.5 ml sonication tubes (Diagnode). Each sonication tube contained about 4 million

cells. Upon 8 cycles of sonication with Pico Bioruptor (30sec ON/30sec OFF), the tubes were spun at 12,000 *g*. Then, Triton X-100 1% final concentration was added to the samples. The supernatant of 6 tubes corresponding to $24 \times 10^6$ cells (1 replicate) was pooled. To each replicate 8 μl of Suz12 antibody (D39F6) was added. After overnight incubation in the cold room, the tubes were spun again at 12000 g. The liquid was transferred to new 2-ml tubes. To do the IP, 40 μl of proteinA magnetic bead enrichment was added to the samples. After 3 h of rotating in the cold room, the beads were cleaned up with Tris–HCl 10 mM. To improve the biotinylation, the beads were treated with Klenow 3′-exo minus, T4 PNK, and dNTPs (NEB) to make 3′-overhangs and remove 3′-phosphates. After that, the beads were treated with terminal deoxynucleotidyl transferase (EP0162) and biotinylated nucleotides (ddUTP and dCTP 1:1, Jena Bioscience). The beads were then washed with IP buffer (Tris–HCl pH 7.5 50 mM, Triton X-100 1%, NP-40 0.5%, EDTA 5 mM), and proteins were eluted with elution buffer (SDS 7.5%, DTT 200 mM) for 15 min at 37°C. Eluted samples were diluted in IP buffer. Then, 100 μl of either prS (LysC) or wt beads were added for the DNA enrichment. Streptavidin beads were washed with three times with SDS washing buffer (Tris–HCl 10 mM pH 8, SDS 1%, NaCl 200 mM, EDTA 1 mM), once with BW2x buffer (Tris–HCl pH 8 10 mM, Triton X-100 0.1%, NaCl 2M, EDTA 1 mM), once with isopropanol 20% in water, and three times with acetonitrile 40% in water. The beads were transferred to PCR tubes using acetonitrile 40%. The supernatant was removed, and the beads were resuspended in 15 μl Ambic 50 mM plus DTT 10 mM final concentration. Then, the samples were incubated at 50°C for 15 min to reduce the disulfide bonds. The cysteines were then alkylated with IAA 20 mM final concentration for 15 min in dark. IAA was neutralized by adding DTT 10 mM final concentration. To digest the proteins, 300 ng LysC (Wako) was added to each sample. After an overnight incubation, the supernatant was transferred to a new PCR tube. Then, 200 ng Trypsin was added to each tube. The digestion continued for 6–8 h. Finally, the peptides were cleaned up using ZipTip pipette tips with 0.6 μl $C_{18}$resin (Merck).

The comparative Suz12 ChIP-SICAP experiment in 2i and serum condition was performed by forward labeling (2i heavy, serum light, 2i medium normal IgG control) and reverse labeling (2i light, serum heavy, 2i medium normal IgG). The labeled cells were separately subjected to the Suz12 ChIP-SICAP protocol. After DNA biotinylated, chromatin fragments were eluted from the IP beads by 100 μl of the elution buffer (SDS 5%, DTT 200 mM) for 15 min at 37°C. Then, three channels were pooled, and the volume was topped up to 1,400 μl by IP buffer. Then, 150 μl of prS beads was added to the samples to capture the chromatin fragments. The rest of the procedure was carried out as described above.

### Lipid pull-down for the identification of sphingolipid-interacting proteins

Lipid pull-down experiments were performed essentially as described before (Gerl *et al*, 2016). In brief, engineered A549(SGPL1 cells were fed with photoactivatable and clickable (pac) sphingosine and labeled with light ($^{12}C_6$ $^{14}N_4$ L-arginine and $^{12}C_6$ $^{14}N_2$ L-lysine), medium ($^{13}C_6$ $^{14}N_4$ L-arginine and $^{12}C_6$ $^{14}N_2$-d4 L-lysine), or heavy ($^{13}C_6$ $^{15}N_4$ L-arginine and $^{13}C_6$ $^{15}N_2$ L-lysine) SILAC amino acids. The metabolized, clickable sphingolipids were linked to biotin through click chemistry following UV irradiation (in light and heavy

channels). The non-UV-irradiated (medium channel) was used as the negative control. Three biological replicates were performed. Cells were lysed in SDS 1% in PBS, and around 270 μg of input material were used per each sample. 600 μl of prS beads or 400 μl of slurry NeutrAvidin beads (Thermo 29200) was conditioned in SDS 0.2% in PBS and then added to the triple SILAC-coded sample. The enrichment was performed o.n. at RT in rotation. Protease-resistant beads were then washed three times with SDS washing buffer, once with NaCl washing buffer (Tris–HCl pH 7.5 10 mM, EDTA 1 mM, Triton X-100 0.1%, NaCl 2M), twice with isopropanol 10% and twice with acetonitrile 20% and then transferred to a PCR tube and resuspended in ammonium bicarbonate (AmBic) 50 mM. NeutrAvidin beads were washed with SDS 1% in PBS 1x, and proteins were eluted in with 50 μl of agarose elution buffer (Tris–HCl pH 6.8 100 mM, SDS 4%, β-mercaptoethanol 4%) and incubated at RT for 1 h followed by 95°C for 30 min. Samples were subjected to protein clean up with SP3 beads as previously described (Hughes et al, 2014) and resuspended in AmBic 50 mM. DTT 7 mM final was added to the samples for disulfide bond reduction at 95°C for 30 min, followed by alkylation with iodoacetamide 12 mM at RT for 40 min in the dark. Reaction was quenched with DTT and proteins were trypsin digested overnight at 37°C. Peptides were cleaned up by SP3 protocol as previously described (Hughes et al, 2014). Samples were injected and analyzed via MS as above-mentioned for BioID samples.

### Plasma membrane proteins

Selective labeling of cell surface presented proteins was performed as described previously (Kalxdorf et al, 2017). In brief: $10 \times 10^6$ HeLa cells were washed with PBS followed by oxidation of carbohydrates with 1 mM sodium metaperiodate (Thermo Fisher Scientific, #20504) in pH 6.5 adjusted PBS at 4°C for 10 min in the dark. Subsequently, cells were washed with PBS followed by biotinylation with 1 mM alkoxyamine-PEG4-biotin (Thermo Fisher Scientific, #26137) in presence of 10 mM aniline (Merck, #242284) for 10 min at 4°C in the dark. Cells were washed with PBS and pelleted at $340 \times g$. Cell pellets were lysed by boiling for 10 min at 95°C in 200 μl of SDS lysis buffer (4% SDS, 60 mM Tris, pH 7.6, 50 mM DTT). Samples were cooled, sonicated on ice once (Bandelin Sonopuls HD 2200) at 50% power output with one burst of 10 s, and lysates were diluted 1:10 in PBS.

0.4 mg of the cell lysates in 1 ml of PBS-SDS1% were treated with 100 μl of the prS (LysC) or wtS. After 1 h rotation at room temperature, the samples were washed with PBS-SDS 1% three times, and with acetonitrile 40% (w/v) three times. After the final wash, the beads were transferred to the PCR tubes using acetonitrile 40%. Then the beads were resuspended in AMBIC 50 mM plus DTT 10 mM. The beads were then incubated at 50°C for 15 min to reduce the disulfide bonds. After that IAA 20 mM (final concentration) was added to the samples to alkylate the bonds. The samples were kept 30 min in dark. To neutralize IAA, DTT 10 mM (final concentration) was added to the samples. To digest the proteins, 300 ng LysC was added to the beads. Following overnight incubation at 37°C, the supernatant of the prS was transferred to a new PCR tube, and 200 ng trypsin was added to the beads. The digestion continued another 8 h. Finally, the peptides were cleaned up using ZipTip pipette tips with 0.6 μl $C_{18}$ resin (Merck).

### Comparing enrichment efficiency between wtS and prS beads

#### Qpcr

150 bp of ActB gene (mus musculus) was amplified using biotinylated primers. The PCR products were purified using PCR purification kit (Qiagen). The concentration of the PCR product was determined using NanoDrop. One microgram PCR product was added to 1 ml PBS-SDS 1%. Then 100 μl of prS or wtS were added to capture biotinylated PCR products. After 1 h of rotating at room temperature, the beads were separated on the magnetic stand. The supernatants were discarded, and the beads were washed three times using PBS-SDS 1%. At the end, the beads were transferred to PCR tubes. The beads were resuspended in 50 μl of PBS-SDS1%. Then 1 μl of proteinase K (20 mg/ml) was added to the samples. After that the beads were incubated at 95°C for 15 min. The supernatant was collected and purified using AMPure XP magnetic beads (1.8×). The recovery of the biotinylated PCR products was measured using real-time PCR relative to the input control.

#### Dot blot

Around $100 \times 10^6$ cells were fully labeled with 5′-ethynyl-2′-deoxyuridine (EdU) for 16 h. Cells were cross-linked with 1% formaldehyde, then permeabilized with 0.25% Triton X-100 in PBS and washed sequentially once with BSA 0.5% in PBS and once with PBS. Biotin azide was linked to EdU through click-chemistry reaction as previously described (Sirbu et al, 2012), then cells were washed again once with BSA 0.5% in PBS and once with PBS. Cells were lysed with SDS 1% in Tris–HCl pH 8.0 50 mM and sonicated for 5 cycles (30sec ON/30sec OFF) with Pico Bioruptor (Diagenode). Upon centrifugation, samples were diluted to 0.5% SDS and split in 20 fractions which were incubated with increasing amounts of either wt beads or prS beads o.n. at +4°C. Upon pull-down, streptavidin beads were recovered on the magnet and 5 μl of the different flow-through were spot on an already activated PVDF membrane. Upon blocking with BSA 5% in PBS-Tween 20 0.1%, the membrane was incubated with anti-biotin antibody (Cell Signaling, #5597) for 1 h at RT. Dot blot was developed with HRP-conjugated anti-rabbit antibody (sc-2357) and images were acquired on a ChemiDoc (Bio-Rad).

#### Western blot

Cells were harvested in RIPA buffer (10 mM Tris–HCl pH 8.0, 140 mM NaCl, 1 mM EDTA, 0.1% SDS), supplemented with 1% Triton X-100 (Triton, Sigma-Aldrich), and cOmplete protease inhibitors (Roche). The cell lysates were sonicated and spun 12000 g in 10 min. Then, the supernatant was transferred to a new tube, and protein concentration was measured by BCA assay. Protein extracts were separated by SDS–PAGE followed by electrotransfer to a PVDF membrane (Trans-Blot Turbo Transfer, Bio-Rad). The membranes were blocked in PBS+ 0.1% Tween 20 (Bio-Rad) + 5% non-fat dry milk and then incubated overnight at 4°C with primary antibodies: Aebp2 (#14129, 1:1,000, Cell Signaling Technology) and GAPDH (ab181602, 1:5,000, Abcam), Primary antibodies were diluted in PBS + 0.1% Tween + 5% BSA. After several washes in PBS + 0.1% Tween, membranes were incubated with anti-rabbit secondary HRP-conjugated antibody (1:5,000, Cell Signaling Technology 7074). Protein signals were detected by the ECL™ Western Blotting Detection Reagents (GE Healthcare) and Amersham Imager 680 blot and gel imager. Densitometry analysis for the bands was performed using Fiji software (Schindelin et al, 2012).

### Data analysis and data visualization

RAW data were processed with MaxQuant (1.5.2.8, 1.6.2.6) (Cox & Mann, 2008) using default settings. MSMS spectra were searched against the UniProt databases (Human for BioID and Lipid-BioID experiment, mouse for ChIP-SICAP) concatenated to a database containing protein sequences of contaminants. Enzyme specificity was set to trypsin/P, allowing a maximum of two missed cleavages. Cysteine carbamidomethylation was set as fixed modification, while methionine oxidation and protein N-terminal acetylation were used as variable modifications. Global false discovery rate for both protein and peptides was set to 1%. The match-between-runs and re-quantify options were enabled. Intensity-based quantification options (iBAQ and LFQ) were calculated. Perseus free software was used for data visualization (Tyanova *et al*, 2016); after canonical filtering (reverse, potential contaminants, and proteins only identified by site), only proteins with at least 1 unique peptide in all the replicates were considered as identified while only proteins with LFQ or SILAC ratio in all the replicates were defined as quantified. Streptavidin intensity percentage was calculated as the intensity of streptavidin protein over the intensity of all identified proteins (expressed as percentage). For GO analysis, the free online tool g.Profiler was used (Raudvere *et al*, 2019).

## Data availability

The mass spectrometry proteomics data have been deposited to the ProteomeXchange Consortium via the PRIDE (https://www.ebi.ac.uk/pride/) (Perez-Riverol *et al*, 2019) partner repository with the dataset identifier PXD016576.

**Expanded View** for this article is available online.

## Acknowledgements
This work was supported in part by the Deutsche Forschungsgemeinschaft (DFG, German Research Foundation)—Projektnummer 331351713—SFB 1324 (to B.B. and J.K.). We thank Itys Comet and Kristian Helin for providing SILAC-labeled ESCs. M.R. was supported by a postdoc fellowship from EMBO (long-term postdoc fellowship 1217-2017) and a postdoc fellowship from European Commission (Marie Curie Standard Individual postdoc fellowship 752075). We gratefully acknowledge H. Christian Eberl for fruitful discussions.

## Author contributions
JK and M-RR designed the research. M-RR, GS, and MK performed the experiments. JB, LF, and BB provided expertise and material. GS and M-RR analyzed the data. GS, M-RR and JK wrote the manuscript with input from all authors.

## Conflict of interest
The authors declare that they have no conflict of interest.

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
