## [Review Process File · Molecular Systems Biology]

Protease-resistant streptavidin for interaction proteomics

ahmoud-reza Rafiee, Gianluca Sigismondo, Mathias Kalxdorf, Britta Brügger, Julien Béthune and Jeroen Krijgsveld.

Review timeline:

Submission date:	4 th December 2019
Editorial Decision:	23 rd January 2020
Revision received:	30 th March 2020
Editorial Decision:	3 rd April 2020
Revision received:	9 th April 2020
Accepted:	14 th April 2020

Editor: Maria Polychronidou

Transaction Report:

1st Editorial Decision

23rd January 2020

Thank you again for submitting your work to Molecular Systems Biology. We have now heard back from the three referees who agreed to evaluate your study. Overall, the reviewers acknowledge that the presented protease-resistant streptavidin is likely to have an impact on the proteomics field. They raise however a series of concerns, which we would ask you to address in a major revision.

Without repeating all the points listed below, the most substantial concerns are the following:

- All three reviewers mention a related study that was published recently (Barshop et al, 2019) and also reports a trypsin-resistant streptavidin. While the reviewers' comments indicate that this previous study would not necessarily preclude the publication of your work in MSB, they indicate that it would be particularly important to include a direct comparison of your approach to that of Barshop et al, and demonstrate that your approach indeed is superior, as you mention in the Discussion.
- Reviewer #1 questions the need to develop a protease-resistant streptavidin. They mention that perhaps the issue can be resolved by using different beads or different amounts of beads and they are concerned that the issue of streptavidin peptide contamination may not be relevant for most BioID protocols. We would ask you to address these points.
- Reviewer #1 thinks that the analyses of the PRC2 interactors seem rather preliminary. We do not think that providing in depth new insights into the biology of the PRC2 complex is necessary considering that the main focus of the manuscript is the methodology. However, since this analysis is used to illustrate the performance of the protease-resistant streptavidin and its advantages for complex experimental design and for obtaining more accurate results, we would encourage you to include some additional data strengthening this part of the study.

All other issues raised by the reviewers would need to be convincingly addressed. As you may already know, our editorial policy allows in principle a single round of major revision so it is

essential to provide responses to the reviewers' comments that are as complete as possible. Please feel free to contact me in case you would like to discuss in further detail any of the issues raised by the reviewers. Given that there is a recently published related study (Barshop et al, 2019), we would recommend submitting your revised work as soon as possible to not compromise its timeliness.

REFEREE REPORTS

Reviewer #1:

Summary

In this manuscript, Rafiee et al. demonstrate that excessive amounts of streptavidin-derived peptides produced via on-bead digestion of NEB streptavidin-coupled paramagnetic particles can hinder mass spectrometry-based protein identification, and outline an approach to chemically modify the streptavidin to mitigate this problem.

Review

1. My first question is: Is this a solution in search of a problem? In other words, while the authors are certainly correct in suggesting that excessive streptavidin peptides can interfere with peptide/protein identification, I wonder how widespread this problem is, and whether it really represents an important issue in the field?

Can this entire issue be avoided by using similar products from other vendors, or different types of streptavidin matrices?

A brief search through the BioID literature reveals that many labs (maybe more than those using the magnetic system?) use a different type of solid phase matrix - namely, non-magnetic streptavidin-sepharose beads.

This type of reagent is also used at apparently much lower amounts than used here - those using the standard streptavidin-sepharose system are generally using 25-50ul bead slurry per sample, whereas the approach described here uses a whopping 200ul bead slurry. (I imagine that this could also get quite cost-prohibitive?) Could this almost 10-fold difference in matrix explain why this particular lab has problems with streptavidin peptide contamination? Streptavidin peptides do not obviously dominate in the other chromatograms that I have seen from BioID experiments. Many of these publications are in high impact journals, suggesting that the use of the non-magnetic bead system can provide high quality data. So, does the strep-sepharose system suffer from the same issue? Or would simply switching systems solve the entire problem? Is the magnetic bead system somehow "better" than the non-magnetic matrix? How?

2. Just as importantly, the authors refer in passing (in a single sentence in the Discussion) to a directly relevant recent publication, "Chemical derivatization of affinity matrices provides protection from tryptic proteolysis" in J. Proteome Research that came out at the end of 2019. While they state that, "This reduction (in streptavidin peptide contamination) is much more profound than in a recently introduced method using different chemistry (Barshop et al 2019).", no real effort has been made to compare the two methods. How does the current approach compare to this recently published method in a head-to-head comparison? Why is your method better?

3. Finally, the authors have also attempted to sandwich a small amount of biology into the manuscript (PRC2 interactors, different isoforms of Aebp2, etc. in ESCs). This work is really not well developed, and does not appear to be backed up by any other complementary type of validation. Publication of these data would require significant additional work, so should probably not be included in this manuscript.

Bottom line

I am satisfied that the approach used here indeed protects streptavidin from trypsinization, and that this can improve mass spectrometry results from the NEB magnetic particles.

I am unsure, however, whether the method is applicable to most published BioID protocols, how much impact this will have on the BioID field, whether the entire issue could be solved by simply switching streptavidin matrices, and how many labs will actually use it. As such, I would suggest that the work would be more appropriate and relevant in a mass spectrometry specialty journal.

Reviewer #2:

The manuscript by Rafiee and colleagues describes a chemical treatment to produce a protease-resistant streptavidin. Lysines and arginines are modified by reductive demethylation and 1,2-cyclohexanedione addition, respectively. The authors demonstrate that the resulting streptavidin still binds biotin and that it is resistant to trypsin and lys-c digestion. The authors go on to use the method in 3 different types of biotinylation-based experiments. Given the exponential growth of experiments using biotin as a handle, this research should have a high impact on the field of proteomics.

My only concern with the manuscript is the treatment of a similar paper that appeared in JPR in September of this year by Barshop et al. This paper also reports the chemical modification of streptavidin with similar results in making it resistant to protease digestion. The authors mention this paper with a single sentence-"this reduction is more profound than in a recently introduced method using different chemistry (Barshop et al, 2019)" Given that these two papers are very similar and that the lysine chemistry is the same (reductive demethylation) this paper should be highlighted in its own paragraph in the introduction.

The authors do a wonderful job of making this fairly technical issue understandable and interesting.

Reviewer #3:

The work by Rafiee and colleagues describes the chemical modification of streptavidin to render this popular capture protein protease resistant. In this way they prevent the generation of important contaminants in mass spectrometry analysis. They show the benefit of this approach in different proteomics applications where biotin-streptavidin capture is utilized: ChIP-SICAP, BioID, surface biotinylation, and analysis of lipid-binding proteins.

The authors provide compelling data that the modified version of streptavidin indeed enhances MS analysis in the different settings. The gain obtained by these 'cleaner' preps is substantial and can impact (expensive) MS run times significantly as shown by the comparisons made. There is no doubt that groups will start applying this clever trick, and that commercial solutions will become available soon.

In the discussion, the authors refer to the Barshop manuscript of last year which pursues exactly the same concept but using different chemistry. These authors also used BioID and surface biotinylation as experimental settings. While Rafiee et al argue why the current approach is superior, it is difficult to assess this now. A side-by-side comparison can be considered here. In addition, it should be noted that the Barshop approach was also applied for other matrices (antibody-based pull down).

Minor comments:

For the BioID experiments, it should be clear that the control cells also express a BirA* construct (which is the case). Please state this clearly in the relevant figures and the figure legends.

Some discussion on the potential use of exclusion lists (as used for trypsin peptides) should be included in the manuscript. Also comment on the absence of streptavidin in the search space (and the implications thereof on the major search engines). A lot of scientists do not add streptavidin to the search space.

Some reports use antibodies in BioID workflows to allow the discovery of biotin-labelled peptides (e.g. BioSITE and PMID:29039416). The degradation of streptavidin by trypsin would imply the release of biotinylated peptides in the sample. Was this observed by the authors? How is this affected (or not) by the chemical modification? This should be assessed/discussed as well.

Response to reviewers

We appreciate the constructive comments by the reviewers, which helped us to make this study more convincing for the scientific community. Below we addressed their concerns in a point-by-point manner.

Reviewer #1:

Summary: In this manuscript, Rafiee et al. demonstrate that excessive amounts of streptavidin-derived peptides produced via on-bead digestion of NEB streptavidin-coupled paramagnetic particles can hinder mass spectrometry-based protein identification, and outline an approach to chemically modify the streptavidin to mitigate this problem.

Review

1. My first question is: Is this a solution in search of a problem? In other words, while the authors are certainly correct in suggesting that excessive streptavidin peptides can interfere with peptide/protein identification, I wonder how widespread this problem is, and whether it really represents an important issue in the field? Can this entire issue be avoided by using similar products from other vendors, or different types of streptavidin matrices?

Response: Streptavidin contamination is a common problem for mass spectrometric analysis of any biotin-based enrichment, however the existence or extent of the problem is usually not reported in the scientific literature. For instance, this includes ourselves when introducing ChIP-SICAP (Rafiee et al, Mol Cell 2016). It may also be possible that streptavidin contamination and its adverse effects go unnoticed when not looking into individual chromatograms, or when omitting streptavidin as a common contaminant during a data base search for the interpretation of MS data. This issue is not dissimilar from antibody contamination in co-IP experiments, where usually only the proteins are reported that come down with the antibody, without speaking about the contamination of antibody-derived peptides in mass spectrometry. Specifically, problems arise with regard to easy overloading of LC columns, blocking of columns, limiting the amount of sample that can be loaded, and impacting peptide intensity-based peptide quantification. Although not frequently reported, these practical issues are well-known to direct operators of mass spectrometry in our own labs, in the core facilities at our respective institutions, and in labs of colleagues around the globe who were eager to test/implement our protocol when hearing about it. Specifically, we have shared the protocol prior to publication with major labs at the DKFZ, Heidelberg University, EMBL Heidelberg, Crick institute, UPenn, UCLA, and Cold Spring Harbour Lab, and we have been highly encouraged by their experience and positive feedback.

To the best of our knowledge, the reported issue is common to all types of streptavidin or neutravidin beads, regardless of the provider and type of beads. In fact it is inherent to the use of a protein (streptavidin here) irrespective of the bead it is coupled to, that is then treated with a protease. Although most of our data were obtained from magnetic beads, we also report that avidin-sepharose beads suffer from the same issue (Figure 3j-k-l).

A brief search through the BioID literature reveals that many labs (maybe more than those using the magnetic system?) use a different type of solid phase matrix - namely, non-magnetic streptavidin-sepharose beads. This type of reagent is also used at apparently much lower amounts than used here - those

using the standard streptavidin-sepharose system are generally using 25-50ul bead slurry per sample, whereas the approach described here uses a whopping 200ul bead slurry. (I imagine that this could also get quite cost-prohibitive?) Could this almost 10-fold difference in matrix explain why this particular lab has problems with streptavidin peptide contamination? Streptavidin peptides do not obviously dominate in the other chromatograms that I have seen from BioID experiments. Many of these publications are in high impact journals, suggesting that the use of the non-magnetic bead system can provide high quality data. So, does the strep-sepharose system suffer from the same issue? Or would simply switching systems solve the entire problem? Is the magnetic bead system somehow "better" than the non-magnetic matrix? How?

Response: Historically, sepharose beads were developed before the magnetic beads, however, magnetic beads are cleaner (i.e. produce less background) and are easier to work with. Yet, we observed massive streptavidin contamination using both bead types.

The amount of the streptavidin beads to be used depends on the type of experiment, the abundance of the biotinylated molecules and the intrinsic capacity of the beads. E.g. this can be done as we showed in Fig 1c where we optimized the amount of beads required to deplete biotinylated biomolecules. In ChIP-SICAP assays we normally use 50ul of the streptavidin magnetic beads. Even with this amount (without chemical derivatization) we observed intense streptavidin peaks that saturate the analytical column, thus both reducing the separation capacity and compromising peptide identification and quantification. Besides, looking into the literature, there is no consensus in the amount of beads that are used, hence the 25-50ul beads mentioned by the reviewer is not a common benchmark. Also, it should be kept in mind that bead-types vary in lot in their capacity (i.e. density of streptavidin on the bead surface, and amount of beads per volume unit). We found the following examples, illustrating the diversity in the amounts of beads that are used:

- Liu X et al Nat Comm 2018 (DOI: 10.1038/s41467-018-03523-2) 200uL of Strep-tactin beads (IBA, GmbH)
- Schopp I et al Nat Comm 2017 (DOI: 10.1038/ncomms15690) 200uL of Dynabeads MyOne Streptavidin C1 (Invitrogen) per 3-3.5mg of sample
- Zhu C. et al Mol Cell 2019 (DOI: 10.1016/j.molcel.2019.06.010) 200uL of PBS-washed MyOne Streptavidin T1 Dynabeads (Life Technology)
- Barshop et al JPR 2019 (DOI: 10.1021/acs.jproteome.9b00254) 125uL of slurry beads

It is important to note that after modification of lysines and arginines of streptavidin, there is no limitation in the amount of beads that can be used since contamination is prevented. Hence, depending on the assays one can apply sufficient beads to capture almost all the biotinylated biomolecules.

2. Just as importantly, the authors refer in passing (in a single sentence in the Discussion) to a directly relevant recent publication, "Chemical derivatization of affinity matrices provides protection from tryptic proteolysis" in J. Proteome Research that came out at the end of 2019. While they state that, "This reduction (in streptavidin peptide contamination) is much more profound than in a recently introduced method using different chemistry (Barshop et al 2019).", no real effort has been made to compare the two methods. How does the current approach compare to this recently published method in a head-to-head comparison? Why is your method better?

Response: This is absolutely a valid point and we apologize for the omission. In the revised manuscript we have now performed two side-by-side comparisons between our protocol and Barshop et. al. protocol, showing 3 main advantages of our approach

1. Shown in a new Supplementary Fig. S2c, we compared mock beads, derivatized by either method but without loading a biotinylated sample. This demonstrates that our protocol performs for better in preventing streptavidin peptides, in particular those at 354, 655 and 1017 m/z.
2. In a new Supplementary Fig. S2d we compared the beads in a ChIP-SICAP experiment. The streptavidin contamination is at the level of $1E10$ using Barshop's protocol, and at $5E7$ when using our protocol, i.e. a 200-fold difference. Moreover, we identified 25% more proteins using our beads. We also added these data in Appendix table S4.
3. Furthermore, Barshop's protocol to modify streptavidin takes 42 hours (assuming overnight incubation is at least 12 hours), while our protocol takes only 6 hours to complete.

In conclusion, preparation of our beads is easier and faster, and leads to better performance as evidenced by less streptavidin contamination.

3. Finally, the authors have also attempted to sandwich a small amount of biology into the manuscript (PRC2 interactors, different isoforms of Aebp2, etc. in ESCs). This work is really not well developed, and does not appear to be backed up by any other complementary type of validation. Publication of these data would require significant additional work, so should probably not be included in this manuscript.

Response: We thank the reviewer's comment. In the revised manuscript we supplemented the MS data with Aebp2 western blot (new Fig S2g) to support our observation that Aebp2 isoform 3/4 is up-regulated in 2i condition of mES cells. Moreover, these data are in agreement to show that Aebp2 is subject to more elaborate processing than previously thought. However, as the reviewer mentioned, further investigation into the function of Aebp2 isoforms 3/4 in the ground state of pluripotency is beyond the scope of this study.

Bottom line: I am satisfied that the approach used here indeed protects streptavidin from trypsinization, and that this can improve mass spectrometry results from the NEB magnetic particles. I am unsure, however, whether the method is applicable to most published BioID protocols, how much impact this will have on the BioID field, whether the entire issue could be solved by simply switching streptavidin matrices, and how many labs will actually use it. As such, I would suggest that the work would be more appropriate and relevant in a mass spectrometry specialty journal.

Response: We strongly believe our protocol is useful for the scientific community using biotin-streptavidin-based protein isolation and enrichment, which includes but also goes far beyond BioID. For instance, the concept has been essential to develop an improved version of our ChIP-SICAP methodology (Figure 2). The step by step protocol is now available in the following database:

<https://www.protocols.io/private/B6386611536311EA912E0242AC110006>

In addition, the method has already gained popularity in the core facilities of our institutions, and we have received positive feedback from various labs around

the globe with whom we have shared the protocol pre-publication. We thus believe that this manuscript will appeal to the community interested in protein network biology, and that MSB is an appropriate platform to reach this audience.

Reviewer #2:

The manuscript by Rafiee and colleagues describes a chemical treatment to produce a protease-resistant streptavidin. Lysines and arginines are modified by reductive demethylation and 1,2-cyclohexanedione addition, respectively. The authors demonstrate that the resulting streptavidin still binds biotin and that it is resistant to trypsin and lys-c digestion. The authors go on to use the method in 3 different types of biotinylation-based experiments. Given the exponential growth of experiments using biotin as a handle, this research should have a high impact on the field of proteomics. My only concern with the manuscript is the treatment of a similar paper that appeared in JPR in September of this year by Barshop et al. This paper also reports the chemical modification of streptavidin with similar results in making it resistant to protease digestion. The authors mention this paper with a single sentence-"this reduction is more profound than in a recently introduced method using different chemistry (Barshop et al, 2019)" Given that these two papers are very similar and that the lysine chemistry is the same (reductive demethylation) this paper should be highlighted in its own paragraph in the introduction. The authors do a wonderful job of making this fairly technical issue understandable and interesting.

Response: We appreciate the positive assessment of our work, and we apologize for the clear shortcoming. In the revised manuscript we have now performed two side-by-side comparisons between our protocol and Barshop et. al. protocol, showing 3 main advantages of our approach:

1. Shown in a new Supplementary Fig. S2c, we compared mock beads, derivatized by either method but without loading a biotinylated sample. This demonstrates that our protocol performs better in preventing streptavidin peptides, in particular those at 354, 655 and 1017 m/z.
2. in a new Supplementary Fig. S2d we compared the beads in a ChIP-SICAP experiment. The streptavidin contamination is at the level of $1E10$ using Barshop's protocol, and at $5E7$ when using our protocol, i.e. a 200-fold difference. Moreover, we identified 25% more proteins using our beads. We also added these data in Appendix table S4.
3. Furthermore, Barshop's protocol to modify streptavidin takes 42 hours (assuming overnight incubation is at least 12 hours), while our protocol takes only 6 hours to complete.

In conclusion, preparation of our beads is easier and faster, and leads to better performance as evidenced by less streptavidin contamination.

We should also mention that we used different chemicals in our protocol both for Lysine and Arginine derivatizations. For Lysine modification, we used sodium cyanoborohydride, while Barshop et.al. used Dimethylamine borane. For Arginine modification, we used cyclohexanedione, while Barshop et.al. used methyl glyoxal.

Reviewer #3:

The work by Rafiee and colleagues describes the chemical modification of streptavidin to render this popular capture protein protease resistant. In this way they prevent the generation of important contaminants in mass spectrometry analysis. They show the benefit of this approach in different proteomics applications where biotin-streptavidin capture is utilized: ChIP-SICAP, BioID, surface biotinylation, and analysis of lipid-binding proteins.

The authors provide compelling data that the modified version of streptavidin indeed enhances MS analysis in the different settings. The gain obtained by these 'cleaner' preps is substantial and can impact (expensive) MS run times significantly as shown by the comparisons made. There is no doubt that groups will start applying this clever trick, and that commercial solutions will become available soon. In the discussion, the authors refer to the Barshop manuscript of last year which pursues exactly the same concept but using different chemistry. These authors also used BioID and surface biotinylation as experimental settings. While Rafiee et al argue why the current approach is superior, it is difficult to assess this now. A side-by-side comparison can be considered here. In addition, it should be noted that the Barshop approach was also applied for other matrices (antibody-based pull down).

Response: We appreciate the positive assessment of our work, and we apologize for the clear shortcoming. In the revised manuscript we have now performed two side-by-side comparisons between our protocol and Barshop et. al. protocol, showing 3 main advantages of our approach:

1. Shown in a new Supplementary Fig. S2c, we compared mock beads, derivatized by either method but without loading a biotinylated sample. This demonstrates that our protocol performs for better in preventing streptavidin peptides, in particular those at 354, 655 and 1017 m/z.
2. in a new Supplementary Fig. S2d we compared the beads in a ChIP-SICAP experiment. The streptavidin contamination is at the level of $1E10$ using Barshop's protocol, and at $5E7$ when using our protocol, i.e. a 200-fold difference. Moreover, we identified 25% more proteins using our beads. We also added these data in Appendix table S4.
3. Furthermore, Barshop's protocol to modify streptavidin takes 42 hours (assuming overnight incubation is at least 12 hours), while our protocol takes only 6 hours to complete.

In conclusion, preparation of our beads is easier and faster, and leads to better performance as evidenced by less streptavidin contamination.

Minor comments: For the BioID experiments, it should be clear that the control cells also express a BirA* construct (which is the case). Please state this clearly in the relevant figures and the figure legends.

Response: Thank you for the point. Fig S3A and the manuscript were revised to indicate this.

Some discussion on the potential use of exclusion lists (as used for trypsin peptides) should be included in the manuscript.

Response: We also tried this solution long time ago. Although the number of identified streptavidin peptides can be reduced this way, it does not alleviate the fundamental issue that streptavidin-derived peptides are still present in the sample, that they limit the amount of sample that can be loaded on-column, that they distort chromatographic performance, and that sampling depth and peptide

quantification are compromised. We have added a statement to this effect in the manuscript (page 2).

Also comment on the absence of streptavidin in the search space (and the implications thereof on the major search engines). A lot of scientists do not add streptavidin to the search space.

Response: Of course, we have included Streptavidin to the data base used for protein identification, with the explicit purpose to identify streptavidin-derived peptides in the sample, and to quantitatively determine their effect the overall outcome of the various applications shown in the manuscript. Omitting streptavidin from the data base would blind the researcher to seeing streptavidin in the eventual list of identified proteins (i.e. computational elimination), however this is not a solution to the underlying contamination problem, exactly as argued above for the use of an exclusion list. Therefore, we want to emphasize that superior performance of protease-resistant streptavidin beads is the result of avoiding contamination in the first place (i.e. physical elimination), and that identical results will be achieved whether or not streptavidin is in the database.

Some reports use antibodies in BioID workflows to allow the discovery of biotin-labelled peptides (e.g. BioSITE and PMID:29039416). The degradation of streptavidin by trypsin would imply the release of biotinylated peptides in the sample. Was this observed by the authors? How is this affected (or not) by the chemical modification? This should be assessed/discussed as well.

Response: We are a bit confused by this point, since by using protease-resistant beads, streptavidin is NOT degraded, and hence biotinylated peptides should NOT end up in the sample after on-bead digested or captured proteins. That said, it may conceptually be conceivable that our prS beads could be used to identify biotinylated peptides in another way, e.g. in a 2-step process where first proteins that are captured are digested on beads to liberate 'regular' peptides. Next, biotinylated peptides that have remained bound to streptavidin may then be eluted for MS analysis. However, this will require very harsh conditions to break the very strong bond. We have not explored this, and we imagine that this will require significant optimization to get this to work with sufficient sensitivity to reach acceptable coverage of modified peptides. Yet, this may be an interesting future application of prS beads.

2nd Editorial Decision

3rd April 2020

Thank you for sending us your revised manuscript. We think that the additional analyses, clarifications and direct comparisons of your approach to that of Barshop et al, 2019 satisfactorily address the issues raised by the reviewers. As such, I am glad to inform you that your manuscript is now suitable for publication, pending some minor editorial issues listed below.

2nd Revision - authors' response

9th April 2020

The Authors have made the requested editorial changes.

Accepted

14th April 2020

Thank you again for sending us your revised manuscript. We are now satisfied with the modifications made and I am pleased to inform you that your paper has been accepted for publication.

Corresponding Author Name: Prof. Jeroen Krijgsveld

Journal Submitted to: MSB

Manuscript Number: MSB-19-9370RR